

# Genetic analysis of two species of Mnomen in the Kalamazoo Watershed reveal panmixia in *Z. Aquatica*, structure among *Z. Palustris*, and hybridization in areas of sympatry

Katarina A. Kieleczawa[1,*], Kaylee Luke[2,3,*] and Andrew Gregory[1]

[1] Department of Biology, University of North Texas, Denton, TX, United States of America
[2] Bowling Green State University, Bowling Green, United States of America
[3] LabCorp, Colombus, United States of America
[*] These authors contributed equally to this work.

Corresponding author
Andrew Gregory,
andrew.gregory@unt.edu

## ABSTRACT

Mnomen or wild rice of the genus *Zizania* is an important part of Native American culture, especially in Michigan for the Ojibwe nation. An oil spill in 2010 along the Kalamazoo River and the subsequent clean-up lead to renewed interest in management of Mnomen within the Kalamazoo watershed. The affected water bodies were surveyed for *Zizania* species to map existing populations, determine the existing genetic diversity and species present, and to identify potential hybridization. Using Traditional Ecological Knowledge of rice beds and opportunistic sampling of encountered plants, 28 rice beds were sampled. Two species of *Zizania* were identified *Z. palustris* and *Z. aquatica*. Genetic diversity was measured using 11 microsatellite loci and was moderately high for both species (*Z. aquatica* $H_E = 0.669$, $H_0 = 0.672$, $n = 26$ and *Z. palustris* $H_E = 0.697$, $H_0 = 0.636$, $n = 57$). No evidence of population bottle-necking was found. *Z. palustris* was found to have $k = 3$ populations on the landscape, while *Z. aquatica* was found to be a single panmictic population. Several individual hybrids were confirmed using genotyping and they were all found in areas where the two species co-occurred. Additionally, *Z. aquatica* was found to have expanded into areas historically with only *Z. palustris* downstream of the oil spill, potentially due to dredging and sediment relocation as part of the clean-up effort.

# INTRODUCTION

Wild rice of the genus *Zizania,* or Mnomen as it is known to indigenous peoples, is both ecologically important to the freshwater ecosystems of Kalamazoo watershed of Michigan and culturally important to native peoples of the Ojibwe Nation. In July of 2010, the Kalamazoo River suffered major damage when a pipeline operated by Enbridge leaked over one million gallons of oil into the surrounding waterways and soil (*EPA, 2017*). Thirty-five

**How to cite this article** Kieleczawa KA, Luke K, Gregory A. 2023. Genetic analysis of two species of Mnomen in the Kalamazoo Watershed reveal panmixia in *Z. Aquatica*, structure among *Z. Palustris*, and hybridization in areas of sympatry. *PeerJ* 11:e15971
http://doi.org/10.7717/peerj.15971

miles of the Kalamazoo River and its tributaries were closed for clean up until June of 2012 (*EPA, 2017*). Costs to clean up the river and tributaries would eventually exceed $1.21 billion (*EPA, 2017*). Following clean-up of the oil spill, there was interest in the degree to which the oil spill and subsequent clean-up activities may have impacted *Zizania* populations along the Kalamazoo River and its tributaries.

Prior to the oil spill and subsequent clean-up, there were three distinct sub-species of *Zizania* found throughout this system. *Zizania palustris* is nationally rare, but is commonly found throughout the Great Lakes region of the US. *Zizania palustris interior* is closely related to *Z. palustris*, but has recently been recognized as a distinct species. Lastly, *Z. aquatica* is a riverine or coastal rice species commonly found along large rivers and coastal estuaries of the United States, including a majority of our sample locations in Michigan (*Ford-Lloyd, Newbury & Virk, 2001*). Wild rice has shown strong evidence for co-adaption in geographically distinct aquascapes that have resulted in co-adaptive gene complexes associated with agronomically important quantitative traits (*Jump, Marchnat & Penuelas, 2009*).

In 2003, the Anishinaabe people of the Great Lakes region, created an Integrated Natural Resource Management Plan (INRMP) which set goals for the management of cultural and ecologically important resources found on tribal controlled lands (KB-1152-2003). The Wild Rice and Native Plants component of the plan set fourth four key goals relating to native plants, such as Mnomen. These goals centered around the management of the native plants and encouraging increased cultural use. The goals for wild rice include mapping and monitoring populations along natural waterways, as well as monitoring and maintaining adequate adaptive potential (KB-1152-2003). One key knowledge gap noted by the INRMP was the distribution and degree to which all three species co-occurred, or not throughout the Kalamazoo Riverine system.

The goals of this project were to address key knowledge gaps identified by the INRMP. Specifically, we sought to: (1) map the distribution of each of the three species of wild rice along reaches of the Kalamazoo River and tributaries, and specifically focused on areas putatively impacted by the 2010 Enbridge oil spill and subsequent clean-up; (2) assess the existing genetic diversity and structure of wild rice of the Kalamazoo watershed following the oil spill and clean up, and (3) although not specifically expected prior to the start of this study, we also sought to measure rates of hybridization and introgression among extant wild rice along impacted reaches of the Kalamazoo River and its tributaries.

## MATERIALS & METHODS

Prior to the Enbridge spill in 2010, there were limited to no *official* records of Mnomen along the main channel of the Kalamazoo River. However, the Anishinaabe regularly harvested *Palustris* from the Kalamazoo River along the western edge of our study site (Lee Sprig, Anishinaabe Mnomen expert, Pers. comm., 2017–2019 Fig. 1). Following river clean-up, rice beds were also noted east of Albion College—there were anecdotal reports that these beds may have been intentionally transplanted to these locations by some local peoples. Additionally, *Aquatica* was the dominant and possibly only Mnomen species found
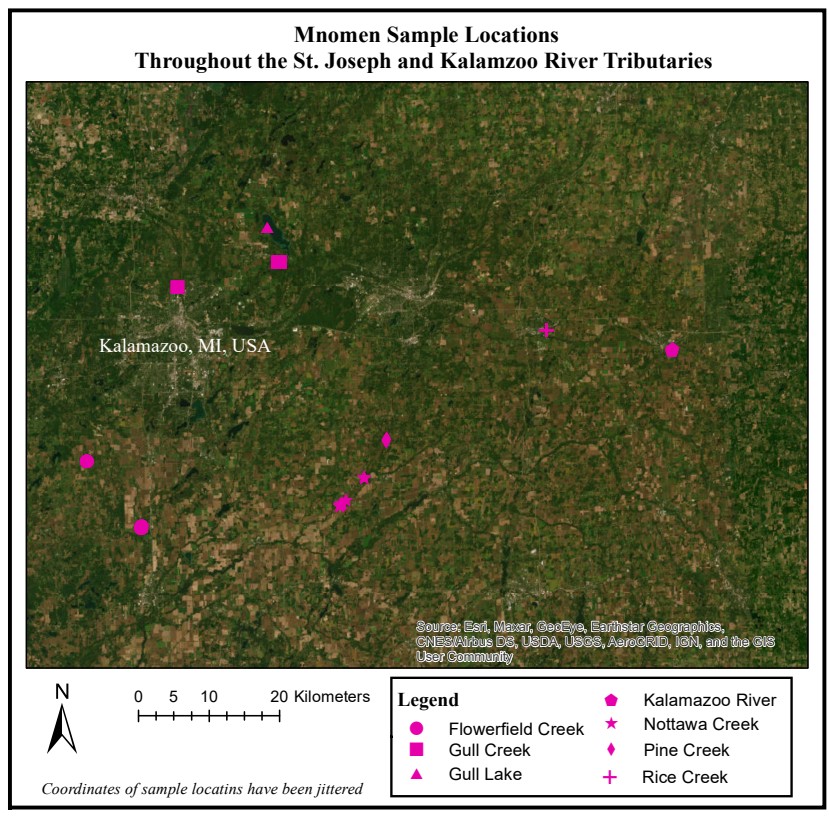

**Figure 1** **Mnomen sample locations throughout the St. Joseph and Kalamazoo River Tributaries.** Map of 28 locations (rice beds) that were sampled throughout the Kalamazoo and St.Joseph River Watershed.

along Nottawaseppi Creek prior to the 2010 Enbridge spill. Anishinaabe rice harvesters began to note the occurrence, or an increased occurrence, of both *Palustris* and *Aquatica* along Nottawaseppi Creek following clean-up (Lee Sprig, Anishinaabe Mnomen expert, Pers. Comm., 2017–2019). We were directed and accompanied to these various rice beds by local rice harvesters, and tribal members assisted during field surveys.

Prior to the start of field work, 11 locations were identified for sampling based on Traditional Ecological Knowledge of culturally important wild rice beds throughout this system, and access to all sites was made possible by the cooperation of members of the Nottawaseppi Huron Band of the Potawatomi *via* ceded territories clause of the Keweenaw Bay, Integrated Resource Management Plan of 1936 as amended in 2002–2012. While traveling to the *a priori* defined sample locations we recorded the location of additional rice beds located, and opportunistically sampled them as well. Using this approach, we sampled 28 beds for a total of 113 rice samples (30 *Palustris* and 80 *Aquatica*; Fig. 1).

At each sample location, we collected an ~2 cm clipping of wild rice leaf tissue and stored it on dry ice until extraction could take place back at the lab. At each sample location we also collected covariate data including coordinates, water temperature, water pH, and putative species identity of rice being collected. *In situ* species identification was carried
out by local indigenous tribal members that are known experts on wild rice identification. Field identification of wild rice was later confirmed using genetic analysis.

Prior to chemical extraction of DNA, all samples were cold shattered by first storing them at −20 °C for 24–72 h, and then mixed with liquid nitrogen and agitated with a Vortex Genie at high speed for 4 h. Samples were then extracted using the EZNA SP Plant DNA kit protocol from Omega BioTek (Omega Bio-tek Inc., Norcross, GA, USA). *Zizania* samples were screened at 19 polymorphic microsatellite primers that have previously been cross-amplified for *Zizania sp.* (*Richards et al., 2004*; *Quan et al., 2009*; Table 1). Each forward primer was tagged with an M13 universal primer attached to the 5′ end (*Schuelke, 2000*). We used Kapa 2G Fast HotStart ReadyMix (MilliporeSigma, St. Louis, MO, USA) for PCR assays following manufacturer protocols with the following modifications: each PCR cocktail reaction consisted of the following: 6.25 μl Kapa 2G Fast HotStart ReadyMix, 2.2 μl water, 0.62 μl forward primer, 0.62 μl reverse primer, 0.41 μl M13 primer, 0.7 μl betaine, 1.5 μl DNA (at∼41 ng/μl concentration), for a total volume of 12.3 μl per reaction. Conditions for PCR amplification were as follows: initial denaturation cycle at 95 °C (3 min), then 35 cycles (denaturation, annealing, extension) at 95 °C (15 s)/60 °C (15 s)/72 °C (15 s), followed by one cycle of final extension at 72 °C (3 min), plus an additional 8 cycles of 95 °C (30 s)/53 °C (45 s)/° C 68 °C (45 s), followed by a final extension cycle at 72 °C (3 min) to incorporate the M13 universal primer sequence into the final PCR products. DNA fragments were visualized using the University of Maine DNA Sequencing Facility, Orono ME on an ABI-3730 with Liz-500 size standard.

Multi-locus genotypic data were scored and collated for each individual using GeneMarker 1.97 (SoftGenetics LLC, State College, PA, USA) automated genotypic software (SoftGenetics LLC, State College, PA, USA). The genotypes of each individual were scored manually by three lab technicians and a consensus genotype is used for each individual's genotype. We initially screened 19 microsatellite markers for use with wild rice in this system. Three markers proved to be monomorphic or did not amplify and were eliminated (Table 1). Examination of allele frequencies for the remaining 16 markers indicated the potential for heteroduplexing in two markers (*Dieffenbach & Dveksler, 2003*). As a result of heteroduplexing some markers will have higher-than-expected ploidy (*Qiu et al., 2001*). Conversely, in rare occasions higher than expected ploidy can also result from plant selfing as opposed to biochemistry anachronisms of the PCR reaction. In either case, such markers are generally not suitable for use in standard genotypic analysis of population diversity and structure (*Qiu et al., 2001*). We also tested for deviations from Hardy-Weinberg Equilibrium, using Program GeneAlEx 6.502 (*Peakall & Smouse, 2012*), for Linkage Disequilibrium using GenePop on the Web (*Rousset, 2008*), and for the occurrence of null alleles using Program Microchecker (*Oosterhout et al., 2004*). These analyses found an additional three markers that were either out of Hardy Weinberg Equilibrium, not at Linkage Disequilibrium or had significant occurrence of null alleles (Table 2). Elimination of these markers left us with 11 microsatellite loci suitable for use in this analysis. After eliminating problematic markers, we also eliminated samples with more than 15% missing data to ensure accurate estimates of gene flow and structure, leaving us with 86 individual samples (29 *Z. palustris* and 57 *Z. aquatica*).

**Table 1** Polymorphic microsatellite primer sequences.

| Locus | Primer sequence (5′–3′) | Repeat Motif | Locus size range (bp) |
|---|---|---|---|
| Zt-1 | GCAAATCTCCTGTCTTTTTCT<br>GTTTAGCCAGCTCCCAATGTA | TAGA | 259–267 |
| Zt-13 | ACGTCGTCGTCTTCCTCC<br>GCATATAATTCCGCGTGAAC | TC | 206–250 |
| Zt-18 | CACCATGTCCTGCAATTC<br>TGCACTAGCTCCCTGAAA | TC | 98–114 |
| Zt-22 | CAACCCCAGAAAAACTAAATC<br>TCCAATCTCTCCACCTACAA | AG | 200–230 |
| Zm-25 | GTTCTGAGTTGCAACCTGGT<br>CCCATATGTCAGCGAGACAT | $(CA)_8(TA)_3$ | 118–148 |
| Zm-26 | CGAACCCTGCATCAAACACT<br>GATTCGGGAGTCTCCTAGTT | $(AG)_{12}$ | 162–170 |
| Zm-30 | GCAAGTGGCTGAAGCAAAC<br>CATGACAATCAATGGTACGT | $(AC)_{23}$,<br>$(AT)_9$ | 152–260 |
| Zm-35 | GACTGATGACAACTGATGGA<br>GCACATGCTTGTGTACTTGT | $(GA)_{22}$ | 190–230 |
| Zm-36 | CACGGTCTGTATCGCTTCT<br>GAGAATGTCTAGACGAGAGT | $(AG)_{13}$ | 210–240 |
| Zm-40 | CAAGCAGCAAATAGCTAGCT<br>GCCTTCATCATCTACTATAC | $(CA)_{22}$,<br>$(TA)_{10}$ | 190–230 |
| Zm-44 | TGCGTGATCTTCAATTCCAA<br>GCTACCGAAGATGTCGTTG | $(GA)_{32}$ | 132–154 |
| *Markers screened for use but excluded from final analysis* | | | |
| Zm-5* | GAGCATTCTCCTCAGATAGT<br>CCCTCTGTTTTCGAGATGG | $(AC)_{13}$ | 146–158 |
| Zm-11× | GGAAGCAAAAAAGTGTAATTTG<br>CAAATGCCTGCACGTGGGA | $(TA)_{11}$, $(TA)_{13}$<br>$(TA)_9$, $(AC)_{32}$ | 168–406 |
| Zm-13* | GCACTGTGTACATGTCAAAC<br>GTGAACCTGTCCTTCAGCAA | $(AG)_{11}$ | 200–204 |
| Zt1-16* | GATGAGCAAGCATCTCTGTG<br>GGATGGATGGATGAACTAGG | TC | 141–145 |
| Zm-16× | CTCCTACACATCAAGGATCA<br>AAGTGATGACATTGGCACGT | $(AC)_{15}$ | 228–238 |
| Zt-21* | CTAGCTTGTTCAGACAAATGTT<br>GACTCTGCTGCATCATATCA | TC | 179–198 |
| Zt-23* | GGACGTTGACATTTTCACA<br>GGATCAGTAAATCCAAATCTGT | AG | 250–284 |
| Zm-39× | CAGTCAAGCTCAGCTTGCT<br>GCCTGTTCTCACCACTTGA | $(AG)_8$ | 188–192 |

**Notes.**
Markers denoted with asterisk (*) were eliminated from the analysis due to linkage disequilibrium. Markers denoted with ×
were eliminated due to poor amplification across a large percent of samples.

**Table 2  Genetic Diversity of *Z. palustris* and *Z. aquatica* by loci.**

| | | | | | | Allele Frequency Table | | | | | | | |
|---|---|---|---|---|---|---|---|---|---|---|---|---|---|
| Pop | PI Sibs | Locus | AR | Ho | He | HWE | Pop | PI Sibs | Locus | AR | Ho | He | HWE |
| AQUATICA | 0.0001 | zm30 | 8 | 1 | 0.653 | *** | PALUSTRIS | 0.000081 | zm30 | 8 | 1 | 0.626 | NS |
| | | zt18 | 4 | 0.145 | 0.226 | NS | | | zt18 | 4 | 0.107 | 0.279 | ** |
| | | zm26 | 2 | 0.982 | 0.499 | *** | | | zm26 | 3 | 0.759 | 0.654 | *** |
| | | zt18 | 24 | 0.727 | 0.896 | *** | | | zt18 | 17 | 0.724 | 0.828 | *** |
| | | zm44 | 3 | 0.232 | 0.26 | NS | | | zm44 | 3 | 0.179 | 0.493 | *** |
| | | zm36 | 16 | 0.962 | 0.646 | * | | | zm36 | 9 | 0.8 | 0.613 | *** |
| | | zt22 | 25 | 0.704 | 0.893 | *** | | | zt22 | 19 | 0.5 | 0.915 | *** |
| | | zt13 | 6 | 0.314 | 0.755 | *** | | | zt13 | 8 | 0.455 | 0.742 | *** |
| | | zm25 | 18 | 0.686 | 0.797 | *** | | | zm25 | 11 | 0.741 | 0.764 | *** |
| | | zm40 | 22 | 0.8 | 0.844 | *** | | | zm40 | 16 | 0.857 | 0.85 | *** |
| | | zm35 | 22 | 0.839 | 0.885 | *** | | | zm35 | 19 | 0.875 | 0.908 | * |

**Notes.**

PI Sibs, Probability of one sample genetically being called as another.

AR, Allelic Richness; $H_O$, Observed Heterozygosity; $H_E$, Expected Heterozygosity; HWE, Hardy-Weinberg Equilibrium.

[NS]Not significant.

[*]$P < 0.05$.

[**]$P < 0.01$.

[***]$P < 0.001$.

We used Program GenAlEx 6.502 to estimate genetic diversity (both expected $H_E$ and observed $H_O$) within each species and/or subpopulation of rice within each species. We estimated effective population size ($N_e$) using the Linkage Disequilibrium model (*Hill, 1981*; *Waples, 2006*; *Waples & Do, 2010*), as implemented in NeEstimator V2.1 (*Do et al., 2013*). We used the method proposed by *Jones, Ovenden & Wang (2016)* to calculate confidence intervals by jackknifing over individuals rather than loci to control for pseudoreplication associated with the physical linkage among overlapping pairs of loci. We used a Wilcoxon Sign Rank test of allele frequencies to test for a mode shift in allele frequencies commonly associated with a population bottleneck (*Soulé, 1987*). Additionally, in GenAlEx we calculated allelic richness (AR), and observed and expected heterozygosity $H_O$ and $H_E$ respectively and to test for the Probability of Identity (PI) and the Probability of Identify among siblings (PI$_{sibs}$). PI and PI$_{sibs}$ are generally viewed as measures of the power of the genotypic data set to resolve differences in population structure with a given set of markers.

We used Program Structure 2.2 (*Pritchard, Stephens & Donnelly, 2000*), to test for hybridization between *Z. Palustris* and *Z. aquatica*, and to test for latent population structure within each species (*Porras-Hurtado et al., 2013*). This analysis works by using Bayesian statistics to group individuals into non-mutually exclusive genetic clusters based on their individual genotypes and the individual allele frequencies within a population (*Pritchard, Stephens & Donnelly, 2000*). If an individual is assigned equally to two different clusters it would likely be an F1 hybrid of, in this case, *Z. palustris* and *Z. aquatica* mating (*Hallgren et al., 2003*). Thus, structure can identify individuals that are the result of hybridization between two or more species or regional variants. We also mapped the

location of putative hybrids on the landscape to gain additional insights into the spatial context or environmental conditions which might promote or inhibit hybridization in this system.

Program Structure was also used to test for population structure within each species. If there is substructure present on the landscape it usually leads to violation of HWE, given that the populations are not panmictic. Geneland (*Guillot, Santos & Estoup, 2008*), which uses a similar system to Structure but also incorporates spatially explicit data was used to confirm the results of the Structure clustering analysis.

## RESULTS

Analysis of genetic diversity with GeneAlEx indicated that both species were highly heterozygous and mostly out of Hardy-Weinberg (Table 2). We also had adequate power to identify population structure and characterize genetic diversity; for *Z. aquatica* (PISibs <0.0001; $H_O = 0.672$; $H_E = 0.669$ AR =13.64) and for *Z. palustris* (PISibs <0.000081; $H_O = 0.636$ $H_E = 0.697$ AR =10.64). The 95% CI for $N_e$ for both rice species encompassed infinity, specifically $N_e$ *Z. palustris* = 59.5 (95% CI = 22.8-infinity) and $N_e$ *Z. aquatica* = 663 (95% CI = 114.4-infinity). We similarly detected no significant indication of population genetic bottleneck in *Aquatica* (Wilcoxon Sign Ran Test $P = 0.89697$) nor in *Palustris* (Wilcoxon Sign Ran Test $P = 0.68115$). Moreover, a plot of the allele frequencies within each species exhibits the predicted L-shaped sigmoidal curve associated with a non-bottlenecked and outbred genetic population.

Analysis with Program Structure to differentiate between *Z. aquatica* and *Z. palustris* indicated strongest support for $K = 2$ genetic clusters corresponding strongly to the field identification of each species. Specifically, 5.8% (five out of 86) of samples were misclassified in the field and identified genetically as different rice species. Of these five misidentified samples, four were identified as *Z. aquatica* in the field, but genetically identified as *Z. palustris*, and one was identified as *Z. palustris* in the field, but genetically identified as *Z. aquatica*.

When we plotted the cluster assignment of each individual sample using Sigma Plot along with the 95% confidence interval of cluster assignment, we were able to identify five potential hybrid individuals (Fig. 2). Additionally, four out of five (80%) of samples misclassified in the field were located in areas where genetic hybrids were identified or were growing in rice beds with both species present. This suggests that hybridization may have confounded rice expert's ability to accurately identify some samples.

When we mapped the location of these hybrids, we observe that hybridization tended to occur in areas where both *Z. palustris* and *Z. aquatica* are sympatric. Nottawa Creek shows hybrid activity where *Z. palustris* and *Z. aquatica* beds are close, as well as sections of the Kalamazoo River upstream from where the oil spill occurred (Fig. 3). In Nottawa Creek, it should be noted that wherever hybrids were detected, both *Z. palustris* and *Z. aquatica* were also detected. Downstream of the oil spill in Gull Lake the rice is mostly *Z. palustris,* with the exception of a couple of small beds of *Z. aquatica* that occurred in areas where active sediment dredging and new sediment translocation/restoration had occurred following the oil spill (Fig. 4).

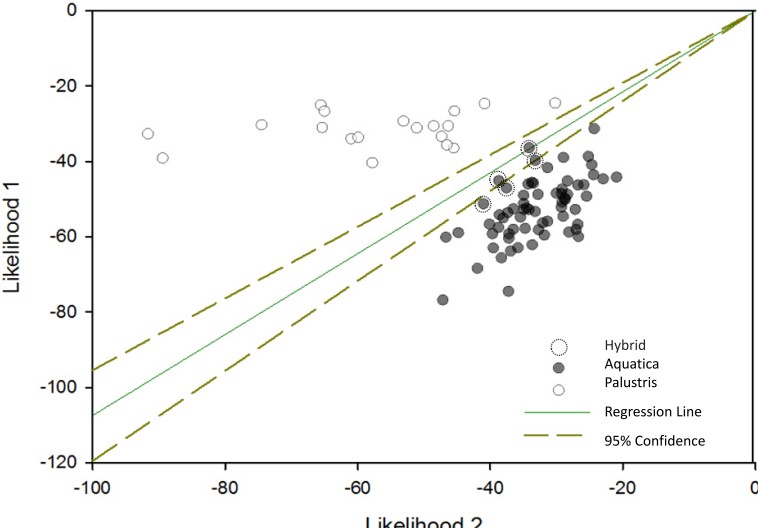

**Figure 2  Palustris *vs* aquatica genotype likelihood.** Negative log likelihood plot of structure results depicting likelihood of each sample membership to each cluster were plotted with a 95% conûdence interval using Sigma Plot 12.5 (Systat Software, Inc., San Jose, CA, USA). This plot shows the relative likelihood that each sample is assigned to cluster 1 or cluster 2. The diagonal transverse with a slope of one indicates an equal likelihood of assignment to either cluster. Individuals falling within the 95% CI of assignment (dotted lines) indicate likely *Z. plaustris × Z. aquatica* hybrids. Individuals to the right of the l x =y transverse, are more likely to be *Z. aquatica*, or are potential F2/F3 hybrid backcrosses into the *Z. aquatica* parental genetic lineage. Individuals left of the $x = y$ transverse are more likely to be *Z. palustris* or F2/F3 backcrosses into the *Z. palustris* lineage.

Using Program Structure hierarchically, we also tested for substructure within *Z. aquatica* and *Z. palustris*. We found equal evidence for $K = 1$ and $K = 3$ potential subpopulations for *Z. palustris* within our study system (Fig. 3), and greatest support for $K = 1$ of *Z. aquatica*, indicting a panmictic population. Analysis with Geneland confirmed greatest support for $K = 3$ genetic subpopulations of *Z. palustris* (Fig. 4).

The three subpopulations of *Z. palustris* tend to be genetically diverse and highly heterozygous. Subpopulation three was the most heterozygous and had almost double the genetic diversity of subpopulations one and two. Measures of genetic differentiation as indicated by $F_{ST}$ values from these three populations show that subpopulation sp1 and sp2 are closely related ($F_{ST}$ sp1 *vs.* sp2 = 0.138; sp2 *vs.* sp3 = 0.141), whereas sp1 *vs.* sp3 are less genetically linked $F_{ST} = 0.066$ (Table 3).

## DISCUSSION

Combining Traditional Ecological Knowledge of the location of Mnomen beds along the Kalamazoo River and its tributaries as well as opportunistic encounters, allowed us to sample and map 28 unique rice beds. Prior to this analysis, no official Mnomen beds were recognized along the Kalamazoo River channel (Lee Sprig and Steve Allen, rice experts for the NHBP Tribe, Pers. comm., 2017–2019). Two species were found, *Zizania palustris*

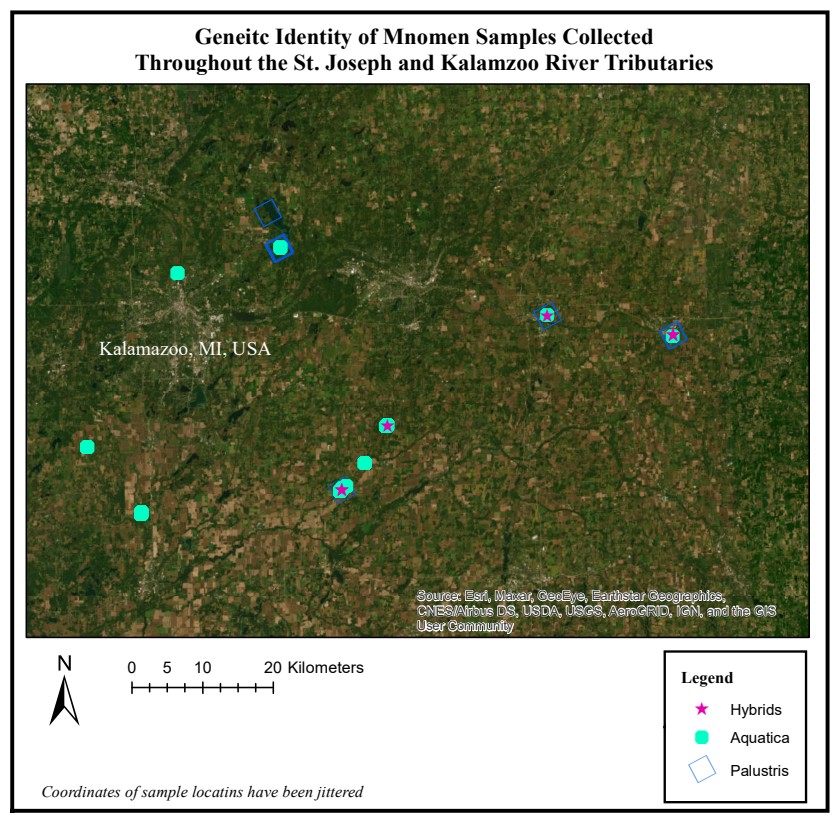

**Figure 3 Genetic identity of mnomen samples collected throughout the St. Joseph and Kalamazoo River Tributaries.** Map of species location and identity based on the genetic species determination. The individuals previously showing evidence of hybridization (yellow stars) occur in areas where both *Z. plaustris* and *Z. aquatica* are sympatric. Nottawa Creek shows hybrid activity where *Z. palustris* and *Z. aquatica* beds are close, as well as sections of the Kalamazoo River above where the oil spill occurred. In Nottawa Creek, it should be noted that wherever hybrids have been detected, both *Z. palustris* and *Z. aquatica* were also detected. These hybrids are mainly F1 generation and some potential F2 hybrids. Below the oil spill in Gull Lake the rice is mostly *Z. palustris* indicating that the river may have been largely *Z. palustris* before the oil spill occurred.

**Table 3 Genetic diversity of *Z. palustris* subpopulations.**

| Allele Frequency Table | | | | | |
|---|---|---|---|---|---|
| **Pop** | **N** | **AR** | **Ho** | **He** | **$F_{IS}$** |
| PA1 | 4 | 3.545 | 0.629 | 0.584 | −0.493 |
| PA2 | 7 | 3.909 | 0.539 | 0.563 | 0.048 |
| PA3 | 18 | 7.727 | 0.672 | 0.628 | −0.07 |

**Notes.**

N, Number of individuals; AR, Allelic richness; Ho, Observed heterozygosity; He, Expected heterozygosity; $F_{IS}$, individual inbreeding coefficients.

($n = 30$) and *Zizania aquatica* ($n = 80$), and genotyped them at 11 microsatellite loci. Dredging also likely resulted in the displacement of *Z. aquatica* to a more southern location

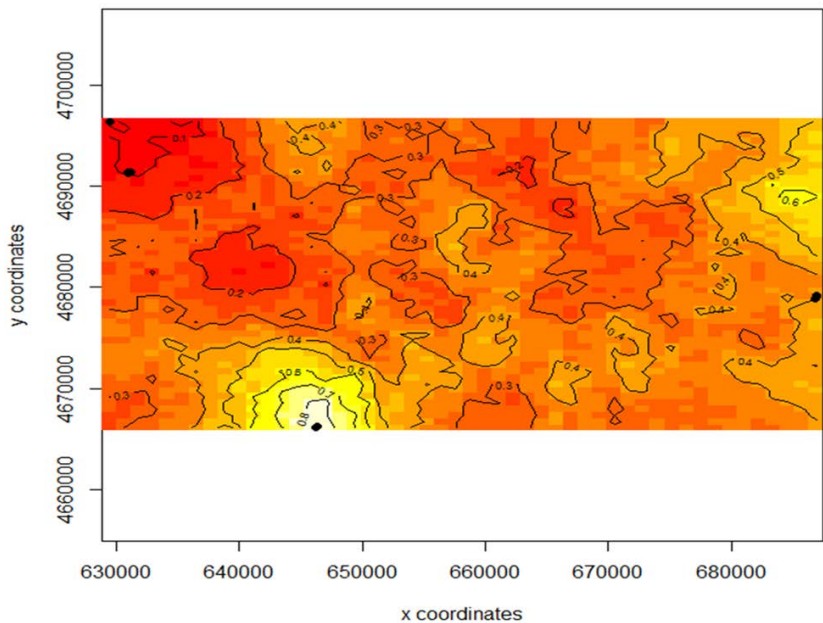

**Figure 4** **Map of posterior probability to belong to cluster 1.** The mapped population clusters from Geneland indicate that physical barriers on the landscape, distance between the beds, or chosen sampling locations, could all be factors in what is driving subpopulation structure within this study system for *Z. palustris*. The contour line map shows the gradient probability of belonging to cluster 1 increasing probability from 0–1; red to white.

than where it was previously suspected of having been present. This indicates that the oil spill may have impacted the presence and distribution of Mnomen in this region.

Of the 86 samples that were genetically analyzed we see a 5.6% misclassification rate, primarily of *Z. palustris*, and most of these were in areas where hybridization is expected based on the presence of both species. Phenotypic plasticity in both species (*Duvall, 1987*) may also have led to the likelihood of misclassification in the field. The hybridization of *Z. palustris* × *Z. aquatica* leads to a significantly reduced fruit set being produced compared to intraspecific crossing (*Duvall, 1987*); however, the offspring are fertile. Unnatural sympatry may lead to reduced yields of wild rice, and would be hard to avoid given the predominance of wind dispersal of pollen for these species.

The anthers of *Zizania sp.* have an independent histology from the flowers and tend to emerge along the lower 1/3 of the 2–3 m long *Zizania sp.* stem (*Zaitchik, LeRoux & Kellogg, 2000*). Wind dispersal of pollen is common across all three Mnomen taxa studied (*USDA, 2001*; *MDNR, 2018*). However, this region is heavily and densely forested. Consequently, Mnomen tends to have a rather limited dispersal ability (*de Wet & Oelke, 1978*; *Zaitchik, LeRoux & Kellogg, 2000*). Therefore, if hybridization among species is to occur, they likely have to be found growing in close proximity to each other. However, hybridization among closely allied lineages is not uncommon among plants when occurring in close proximity to each other (*Whitney et al., 2010*). Our observations of the occurrence of hybrids only in

areas where both species were observed to be growing in adjacent or shared beds further supports the notion that Mnomen has limited dispersal ability. The degree to which hybridization may benefit or negatively impact Mnomen populations of the Kalamazoo warrants further investigation.

*Z. aquatica* exhibits panmixia and evidence was only found for a single population, whereas for *Z. palustris* evidence suggests that three populations exist within this stretch of river. Our results suggest that for *Z. aquatica* gene flow is occurring at a sufficient rate to maintain genetic diversity, though that southern population that was likely displaced by sediment relocation may see differentiation over time if it is not connected to the main populations. The differentiation of *Z. palustris* into three populations suggest that asexual reproduction may be occurring at a greater rate or there is a barrier to gene flow among these populations, such as geographic isolation which agrees with other studies of this species (*Hayes, Stucker & Wandrey, 1989*).

While genetic diversity for both species in this study were moderately high it is possible that not enough stands or smaller stands were sampled. Allelic richness was primarily contributed by larger beds for *Zizania texana*, an endemic and highly geographically restricted wild rice in Texas (*Richards et al., 2007*). Further sampling may be necessary to determine true populations and genetic diversity as well as to differentiate other potential hybrids. Maintaining or enhancing the presence of larger stands may be an important management goal to sustain local genetic diversity. For example, a study done on *Z. latifolia* in China suggested that the rivers did not serve as effective corridors for gene flow among populations (*Chen et al., 2017*). That study showed high inter-population and low intra-population genetic variation was likely the result of inbreeding or clonal breeding as opposed to sexual reproduction necessitated as a result of riverine disturbance resulting in a recent bottleneck (*Chen et al., 2017*). A similar situation may be evolving in the Kalamazoo River and tis tributaries, such that following the oil spill Mnomen has been displaced to more disjunct and fragmented locations, with limited opportunity for genetic mixing. Genetic monitoring and more high-density surveys are needed. Fortunately, given the cultural importance of the Mnomen to the Anishinaabe People, regular high-resolution surveys of this system occur every year during annual rice harvest, this important reservoir of Traditional Ecological Knowledge about the system is available for resource management agencies to use for future conservation and management planning.

## ACKNOWLEDGEMENTS

We thank Lee Sprig and Steve Allen of the Pottawatomie Band of the Ojibwe Nation for assistance in obtaining field site access and identification of Mnomen in the field. We also thank the anonymous members of the Pottawatomie Band of the Ojibwe Nation who assisted us in field sample acquisition. We thank G. Ysassi, H. Olenick, and A. Nichter who were technicians on this project.

### Funding

This project was funded by a grant from the Bureau of Indian Affairs to Andrew Gregory and the Nottawaseppi Huron Band of Potawatomi Nation. The funders had no role in study design, data collection and analysis, decision to publish, or preparation of the manuscript.

### Grant Disclosures

The following grant information was disclosed by the authors:
Bureau of Indian Affairs to Andrew Gregory.
Nottawaseppi Huron Band of Potawatomi Nation.

### Competing Interests

The authors declare there are no competing interests. Andrew Gregory is an Academic Editor for PeerJ. Kaylee Root is currently employed by LabCorp, a diagnostic medical lab testing company not affiliated with this research.

### Author Contributions

- Katarina A. Kieleczawa analyzed the data, prepared figures and/or tables, authored or reviewed drafts of the article, and approved the final draft.
- Kaylee Luke conceived and designed the experiments, performed the experiments, analyzed the data, prepared figures and/or tables, and approved the final draft.
- Andrew Gregory conceived and designed the experiments, performed the experiments, analyzed the data, prepared figures and/or tables, authored or reviewed drafts of the article, wrote the grants that funded the work, and approved the final draft.

### Ethics

The following information was supplied relating to ethical approvals (i.e., approving body and any reference numbers):

This project was conducted under the auspice of an EPA approved Quality Assurance Plan in cooperation with the Michigan Department of Natural Resources.

### Field Study Permissions

The following information was supplied relating to field study approvals (i.e., approving body and any reference numbers):

Nottawaseppi Huron Band of Potawatomi ceded tribal lands.

### Data Availability

The genetic data was made available to PeerJ for peer review and can be requested from the corresponding author Andrew Gregory. The location data will not be made available due to the culturally sensitivity of those data. Maps have been deliberately created at coarse resolution to also help protect the sensitive nature of the sample site locations that are culturally sensitive to and at the request of the Anishinaabe People.

## Supplemental Information

Supplemental information for this article can be found online at http://dx.doi.org/10.7717/peerj.15971#supplemental-information.

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
