# Peer review of "Genetic analysis of two species of Mnomen in the Kalamazoo Watershed reveal panmixia in Z. Aquatica, structure among Z. Palustris, and hybridization in areas of sympatry"

_PeerJ, doi:10.7717/peerj.15971_

## Round 0.1 · original submission · Minor Revisions

Dear authors, please revise the article considering the reviewers' comments.

Reviewer 1 ·

Basic reporting

The manuscript is outstanding and well scripted, although, some correction could be made.
line 27-28: Abbreviation "Ho" is not uniform.
Line 49: "-- -" is out of place symbol and is not useful in manuscript.
Line 50: "foundcommonly" is grammatically wrongly, needed to be corrected.
Line 59: "/" is again out of place symbol and is not useful in manuscript.
Line 92: "Richards et al, 2006" is missing from references.

Experimental design

No Comment

Validity of the findings

No comment

Additional comments

The references contain 15 unused citation which were not refereed in manuscript. these looks dummy references and should be excluded from final m/s.
Line No. 253, 258, 260, 262, 264, 277, 295, 297, 298,327,330, 331, 333, 335 and 343.

·

Basic reporting

1. There are a few spots where sentence structures are not great. Please fix it and rewrite line 112-113 as in a way that is easier to grasp.

4. Line 212 contains a spelling error; I believe the word "lent" should be replaced with the word "led."

Experimental design

NO COMMENT

Validity of the findings

1. Kindly double-check all the references that it should be mentioned in the manuscript also. The manuscript does not include as many references listed in references like in lines 253, 258, 272, 327, 330, and 331.

2. In figure 2 clearly depict which are that five potential hybrids which are mentioned in lines 172-174.

3. Why are only 11 microsatellite markers utilised as more markers can be used? These are very less in a number of markers to validate the results obtained.

Additional comments

Please add additional details to your manuscript as it doesn't seem to be very descriptive overall.

·

Basic reporting

1. Line 212 contains a spelling error; I think that the word "lent" should be replaced with the word "led."
2. There are a few places where sentences and paragraphs are not perfect. Please fix it and rewrite it so that it is easier to understand.

Experimental design

No

Validity of the findings

1. If you have data related to population and manner of hybridization among them before the oil spill. Then add that to make your study more impactful. Because there are many reasons for hybridization and how can you say that only oil spills have done mixing / hybridization among the different population.
2. Verify all the references that need to be included in the manuscript as well. There aren't as many references listed in the manuscript's references section as there are in lines 253, 327, and 330.
3. Kindly add/show GEL images showing 11 polymorphic markers found in study which will give more validation to your study.
4. Figure 3 does not clearly depict hybrids. So, clearly and in relation to lines 179–181, depict hybrids.

Additional comments

Your introduction needs more detail. I suggest that you improve the description at lines 57- 86 to provide more justification for your study (specifically, you should expand upon the knowledge gap being filled).

---

## Round 0.2 · accepted · Accept

Your revised has been approved for publication.